# ART: ACTOR-RELATED TUBELET FOR DETECTING COMPLEX-SHAPED ACTION TUBES

## ABSTRACT

This paper focuses on detecting complex-shaped action tubes in videos. Existing methods assume that actor's position changes slightly in short video clips. These methods either oversimplify the shape of action tubes by representing them as cuboids or conjecture that action tubes can be summarized into a set of learnable positional patterns. However, these solutions may produce an action tube losing the corresponding actor when the actor trajectory becomes complex. This is because these methods rely solely on position information to determine action tubes, lacking the ability to trace the same actor when their movement patterns are intricate. To address this issue, we propose **A**ctor-**r**elated **T**ubelet (**ART**), which incorporates actor-specific information when generating action tubes. Regardless of the complexity of an actor's trajectory, ART ensures that an action tube consistently tracks the same actor, relying on actor-specific cues rather than solely on positional information. To evaluate the effectiveness of ART in handling complex-shaped action tubes, we introduce a dedicated metric that quantifies tube shape complexity. We conduct experiments on three commonly used tube detection datasets: MultiSports, UCF101-24 and JHMDB51-21. ART presents remarkable improvements on all the datasets.

## 1 INTRODUCTION

Spatio-temporal human action detection requires simultaneously localizing an actor and recognizing the action category in a video. Most existing methodologies (Feichtenhofer et al., 2019; Chen et al., 2021; Tang et al., 2020; Pan et al., 2021; Wu et al., 2023; Chen et al., 2023) focus on detecting actions at the frame level, considering temporal information mainly for action recognition rather than actor localization. In this paper, we target a different setting, *i.e* action tube detection at the video level following some earlier works (Kalogeiton et al., 2017;

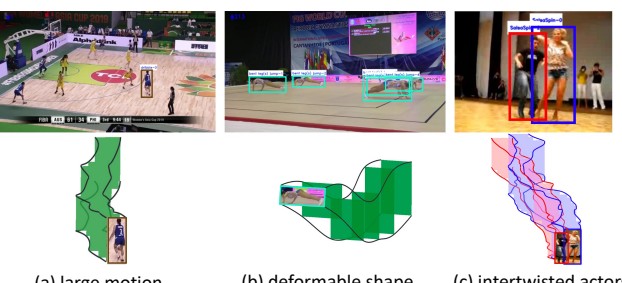

(a) large motion     (b) deformable shape     (c) intertwisted actors

Figure 1: **Complex-shaped tubes from MultiSports (Li et al., 2021) and UCF (Soomro et al., 2012).** Action tubes are extremely complicated in real scenarios due to large motion, deformable shapes, or intertwisted actors.

Hou et al., 2017; Singh et al., 2017). An action tube is defined as a sequence of temporally successive bounding boxes of an actor performing actions (Jain et al., 2014; Kalogeiton et al., 2017). Action tube detection requires temporal consistency for not only action recognition but also for actor localization to generate action tubes. Therefore, it is more challenging than the frame-level setting which only yields independent actor bounding boxes per frame. The goal of this paper is typically designed for **action tube detection**.

The shape of action tubes varies across action categories and types of videos, resulting in some of the actions being easy to detect, while others are challenging. For example, easy actions like brushing hair/teeth, sitting/drinking/clapping, and playing flute/guitar often occur in bounding boxes that are spatially stationary or follow predictable patterns. In contrast, actions involving in sports, dancing, and gymnastics typically come with unpredictable trajectories, which poses challenges to existing action detectors. This is evidenced by the results from previous arts (Feichtenhofer et al., 2019; Chen et al., 2021) when evaluating detection performance on individual categories. Motivated by this, we

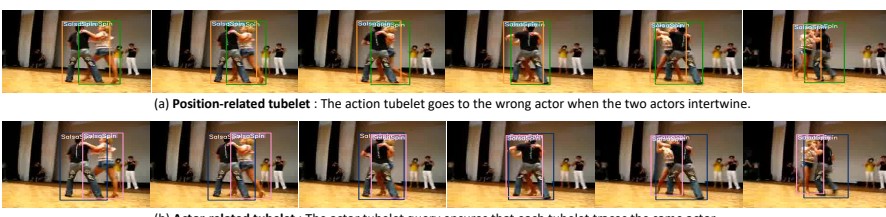

Figure 2: (a) The position-related tubelet fails to generate tubes with complex shapes because it relies solely on position information without considering actor information. (b) Regardless of the complexity of an actor's trajectory, the actor-related tubelet is able to trace the same actor and form precise action tubes.

examine several popular action detection datasets and find that earlier datasets (Jhuang et al., 2013; Soomro et al., 2012; Gu et al., 2018) mainly include actions with limited variation in the shape of tubes, whereas challenging cases emerge in more recent dataset (Li et al., 2021). These challenges are largely due to the intricate shapes of action tubes. For instance, in Fig. 1 (a), a basketball player dribbling past opponents exhibits rapid, irregular motion in the bounding boxes. This complexity is often compounded by camera motion. Fig. 1 (b) shows a more fine-grained action, "bent leg jump" in gymnastics, where the body deformation results in the drastic changes in the bounding box shape. Additionally, when multiple actors are intertwined in interactive actions, e.g. Salsa spin (Fig. 1 (c)), their action tubes frequently overlap and cross, greatly increasing the difficulty of tube detection.

The above analysis shows that it is crucial for an action detector to cope with complex-shaped action tubes. Unfortunately, this property is scarce from existing frameworks. Broadly, current frameworks can be categorized into cuboid-based (Singh et al., 2017; Hou et al., 2017; Kalogeiton et al., 2017; Li et al., 2020a) and query-based (Zhao et al., 2022; Gritsenko et al., 2024) approaches. Specifically, cuboid-based methods rely on pre-designed cuboid anchors or duplicate a bounding box along the time dimension to form a regular-shaped tube. The query-based methods TubeR (Zhao et al., 2022) and STAR (Gritsenko et al., 2024) assumes any action tube can be characterized into a set of learnable spatio-temporal positional queries. All of them are established on a ***position-related*** assumption that an actor's position changes very slightly in short video clips. Therefore, these methods are mainly crafted to detect the aforementioned easy action tubes and tend to fail in detecting tubes with complex shapes like in Fig 1. The root cause is that these methods determine action tubes solely based on position information without considering actor information. An illustration is shown in Fig. 2. When two actors are intertwined, the trajectories of the action tubes become complex. Existing methods adhere to positions and still predict regular-shaped tubes even covering wrong actors (Fig. 2(a)). The orange tubelet begins with the man but ends with the woman, while the green tubelet starts with the woman and ends with the man. This position-related tubelet is unable to effectively distinguish between different actors. To address this limitation, we propose the **A**ctor-**r**elated **T**ubelet (**ART**), which incorporates actor-specific information to generate accurate action tubes. As shown in Fig. 2(b), ART consistently adheres to the same actor over time.

As discussed above, when the actor's trajectory becomes complicated, the position-related assumption underlying existing methods are prone to failure. Intuitively, regardless of how intricate the actor's trajectory becomes, once the tube of an actor is determined, it is much easier to determine the evolution of actions. A natural way to achieve this is by applying a person tracker to construct actor tubelets. However, this approach demands extra effort and data for training or fine-tuning the tracker to adapt to the specific domain. Instead, we adopt a query-based detector and leverage its attention mechanism to formulate tubelet queries that automatically track target actors, eliminating the need for an additional linker or tracker. Specifically, we allow the model to briefly analyze the video beforehand, providing it with knowledge of which actors are present. With this information, we incorporate this prior knowledge into the tubelet queries, referred to as actor tubelet queries. These queries enable the model to effectively determine the presence of the target actor in each frame of the video clip, irrespective of changes in the actor's position.

Our model named Actor-related Tubelet (ART), comprises an Actor Decoder responsible for localizing actors within keyframes, and an Action Decoder that generates the final action tubelets. A Tubelet Query Generator is the bridge to connect the two decoders by dynamically constructing actor tubelet queries. Besides, to evaluate ART for complex action tubes, we propose a metric to measure tube complexity and divide datasets into different subsets with different complexity scores. ART shows big gains for most complicated tubes. In summary, our contributions are:

1. **Impact**: We challenge position-related assumptions in existing methods for complex tubes. We propose the first end-to-end actor-related tubelet (ART) detector for complex tubes and a metric for tube complexity.
2. **Design**: ART empowers the model with the ability to precisely query an actor in every frame to automatically form actor tubelets. This breakthrough allows us to effectively follow an actor within a tubelet, regardless of the complexity of the action tube's shape.
3. **Performance**: ART is an end-to-end system without any bells and whistles and achieves remarkable results on MultiSports, UCF101-24 and JHMDB51-21 in terms of video mAPs.

## 2 RELATED WORK

**Traditional action detection.** Spatio-temporal human action detection in video has garnered sustained attention, *e.g* (Cao et al., 2010; Tran & Yuan, 2012; Weinzaepfel et al., 2015; Peng & Schmid, 2016; Girdhar et al., 2018; Sun et al., 2018; Pan et al., 2021). With the advancements in deep neural networks, significant improvements have been made in the field of action detection in video. Some early works (Peng & Schmid, 2016; Saha et al., 2016; Singh et al., 2017) applied 2D convolutional networks to detect actions per-frame, drawing inspiration from object detection. These methods require linking frame-wise predictions to form action tubes. To effectively leverage temporal information, action detection at the tubelet level (Li et al., 2018; Song et al., 2019; Yang et al., 2019; Li et al., 2020a) has gained significant popularity since it was introduced by Jain *et al* (Jain et al., 2014). Hou *et al*(Hou et al., 2017) and Kalogeiton *et al* (Kalogeiton et al., 2017) employed faster-RCNN/SSD detector with 3D cuboid anchors to generate action tubelets. Subsequently, Yang *et al* (Yang et al., 2019) proposed to progressively refine 3D cuboid anchors across time. Li *et al* (Li et al., 2020a) detected tubelet instances by relying on center position hypotheses instead of cuboid anchors. In order to detect large motions, Singh *et al* (Singh et al., 2022) employed an offline person tracker to generate actor tubes and pooled features based on these tubes. The key distinction between ART and the method is that ART is an end-to-end, transformer-based system that automatically generates actor tubelets, eliminating the need for any external tracking systems.

Recently, 3D convolutional networks are widely used for video understanding due to its superior ability to capture temporal information. Gu *et al* (Gu et al., 2018) integrated a 3D convolutional network into a Faster R-CNN detector to enhance the understanding of action categories by learning spatio-temporal features. Derived from this regime, two-stage methods aimed at improving action recognition by using offline person detectors to localize actors were introduced. Feichtenhofer *et al* (Feichtenhofer et al., 2019) designed a slowfast network for this purpose. Tang *et al* (Tang et al., 2020) and Pan *et al* (Pan et al., 2021) explicitly models relations between actors and objects, which favors for action understanding. Singh *et al* (Singh et al., 2022) and Faure *et al* (Faure et al., 2022) employed ensemble models, incorporating either a tracker or pose estimation, respectively. Beyond two-stage methods, Chen *et al* (Chen et al., 2021) proposed an end-to-end single model capable of jointly training actor localization and action classification. These mentioned methods detected actions per-frame, whereas ART is designed for detecting action tubelets at video level.

**Transformer-based video understanding.** Girdhar *et al*(Girdhar et al., 2019) proposed a video action transformer network to improve action recognition by aggregating features from the spatio-temporal context around actors. Fan *et al* (Fan et al., 2021) and Li *et al* (Li et al., 2022) proposed to learn multiscale feature hierarchies with transformer models for video recognition. Additionally, MeMViT (Wu et al., 2022) processed long videos in an online manner by maintaining prior memory to capture long-term context. Recently, a hierarchical Vision Transformer without the bells-and-whistles (Ryali et al., 2023) presents superior performance on multiple vision tasks, including video recognition. In the realm of action detection, Zhao *et al* (Zhao et al., 2022) proposed TubeR for detecting action tubelets in video clips with 3D convolutional backbone and a transformer encoder-decoder. Subsequently, STAR (Gritsenko et al., 2024) presented a purely-transformer based model. Although ART and TubeR/STAR are all query-based detector, ART exhibits fundamental differences. Specifically, TubeR and STAR learned action tubelet queries from randomly initialized positions, which poses limitations when dealing with complex action tubes, as discussed earlier. In contrast, ART constructs tubelet queries that focus on target actors and remain unaffected by actor positioning, allowing ART to effectively manage more intricate action tubes. Other query-based action detection methods including STMixer (Wu et al., 2023) and EVAD(Chen et al., 2023) generated action boxes only at the frame level. ART, however, is designed for the more challenging task of video-level tube detection.

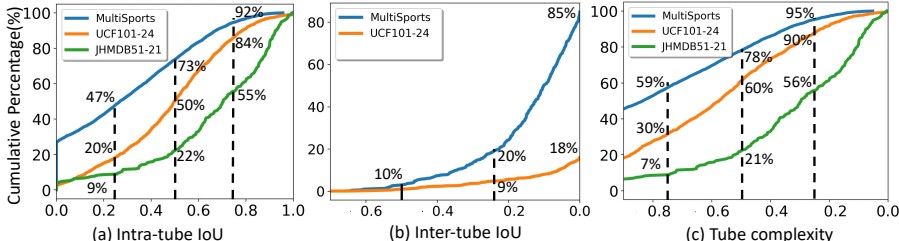

Figure 3: **Cumulative density function of intra-tube IoU, inter-tube IoU and tube complexity.** The intra-tube IoU measures the complexity within a tube and inter-tube IoU measures the complexity of pairs of tube interactions. Tube complexity is calculated from the intra-tube IoU and inter-tube IoU as introduced in Sec. 3.1. The ratios of complex tubes are significant on MultiSports and UCF.

## 3 METHODOLOGY

### 3.1 SHAPE COMPLEXITY OF TUBE

We first introduce a metric designed to quantify the complexity of action tube shapes, enabling us to assess the effectiveness of our method in handling complex scenarios. Designing such a metric is challenging, as it must account for various factors, including camera motion, actor shape deformation, and fast motion. Additionally, interactions between tubes further complicate detection. To address this, we utilize intra-tube IoU (Intersection over Union) to measure the complexity within a tube and inter-tube IoU to assess interactions between tubes within a video.

Given a video containing $M$ tubes $T_1, T_2, ..., T_M$, a tube $T_j = \{B_j^1, B_j^2, ..., B_j^l\}, j \in 1, 2, ..., M, B_j^i$ ($i \in 1, 2, ..., l$) is a bounding box at time $i$. The intra-tube IoU for a tube $T_m$ is the average of IoUs for box pairs in the tube:

$$\text{Intra\_IoU\_m} = \sum_{i=1}^{l-1} \text{IoU}(B_m^i, B_m^{i+1})/(l-1) \tag{1}$$

The lower the Intra_IoU_m, the higher the shape complexity of the tube. Thus, we define the inner complexity of the tube $T_m$ as Intra_C_m:

$$\text{Intra\_C\_m} = 1 - \text{Intra\_IoU\_m} \tag{2}$$

To measure the complexity of $T_m$ due to its interaction with other tubes in the video, we first calculate the inter-tube IoU for $T_m$ and each other tube $T_j$ in the video:

$$\text{Inter\_IoU\_mj} = \text{TIoU}(T_m, T_j), j \in 1, 2, ..., M \& j \neq m \tag{3}$$

TIoU means the tube IoU (Zhao et al., 2022). Tubes that overlap with $T_m$ increase its complexity. To quantify the effect of these overlapping tubes on $T_m$, we normalize all TIoUs between the tubes overlapping with $T_m$ and define the normalized value as an interaction coefficient (Icoe) between $T_m$ and an overlapping tube $T_j$:

$$\text{Icoe\_mj} = \text{Inter\_IoU\_mj}/\sum_j \text{Inter\_IoU\_mj}, j \in 1, 2, ..., M \& j \neq m \tag{4}$$

The interaction complexity of $T_m$ is defined as the weighted sum of inner complexity of the tubes overlapping with $T_m$:

$$\text{Inter\_C\_m} = \sum_{j \neq m} (\text{Icoe\_mj} * \text{Intra\_C\_j}) \tag{5}$$

The final complexity of tube $T_m$ is the sum of inner complexity and interaction complexity:

$$\text{C\_m} = \text{Intra\_C\_m} + \text{Inter\_C\_m} \tag{6}$$

With the defined tube complexity score, we are able to analyze action tube complexity distribution on the datasets to support our motivation. Fig. 3 (a) shows the cumulative density function of intra-tube IoU is plotted for the training sets of MultiSports, UCF and JHMDB. Notably, 30% of the tubes in MultiSports exhibit an IoU of 0.0, indicating significant variation in the bounding boxes within these tubes. JHMDB, which consists of simple actions characterized by short tubes and samll motion, has only 22% of tubes with an IoU lower than 0.5, compared to 50% in UCF and 73% in MultiSports. Fig. 3 (b) shows the cumulative density function of IoU for pairs of tubes (inter-tube) within a video.

A higher overlap between tubes indicates increased complexity in tube interactions. JHMDB is excluded from this plot as it features only a single actor per video. The results demonstrate that 85% of tubes in MultiSports overlap with other tubes, compared to only 18% in UCF. Figure 3 (c) further illustrates tube complexity defined in Eq. 6. In MultiSports, 60% of the tubes have a complexity score greater than 0.8, compared to 30% in UCF and only 7% in JHMDB. In conclusion, both MultiSports and UCF contain a substantial number of complex-shaped action tubes.

In the following sections, we introduce how our ART detects complex-shaped action tubes in a video clip. Fig 4 depicts the whole system of the method.

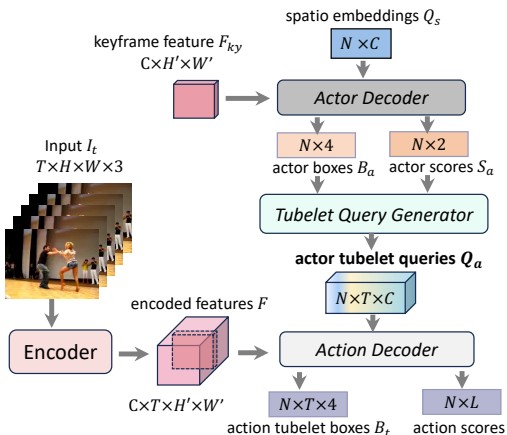

### 3.2 ENCODER

Given a video clip $I_t \in \mathbb{R}^{T \times H \times W \times 3}$ where $T, H, W, 3$ denote the number of frames, height, width, and colour channels, ART first builds an Encoder to extract spatio-temporal video features $F \in \mathbb{R}^{C \times T \times H' \times W'}$, $C$ its latent dimension. Specifically, for a *pure transformer* system, the Encoder adopts a video Transformer network (Ryali et al., 2023) as its backbone to get low-resolution feature maps $F' \in \mathbb{R}^{C' \times T' \times H' \times W'}$. For action tubelet detection, if $T' \neq T$, an interpolation layer will be applied to the temporal dimension and make the new feature maps $F' \in \mathbb{R}^{C' \times T \times H' \times W'}$. A liner layer is further utilized to reduce feature dimension from $C'$ to $C$ and get the encoded video feature $F \in \mathbb{R}^{C \times T \times H' \times W'}$.

Figure 4: Structure of ART. ART takes as input a video clip and extracts video features using Encoder (3.2). It adopts Actor Decoder to detect actors on keyframes and then dynamically builds actor tubelet queries through a Tubelet Query Generator (3.3). Finally, it decodes action tubelets from actor tubelet queries through Action Decoder (3.4).

### 3.3 ACTOR-RELATED TUBELET

#### 3.3.1 PRELIMINARY

A query-based action detection framework learns a set of spatio-temporal positional queries $Q=\{Q^1, ..., Q^n, ..., Q^N\} \in \mathbb{R}^{N \times T \times C}$ to model action tubelet patterns. Here $N$ is the number of queries, $T$ the temporal duration of the tubelet and $C$ the feature dimension. Each query $Q^i = \{Q_1^i, ..., Q_j^i, ..., Q_T^i\}$ contains $T$ spatial positional queries (i.e. $Q_j^i \in \mathbb{R}^C$) corresponding to $T$ frames, respectively. These queries are intended to represent an action tubelet across $T$ frames. Given that the average number of spatial positional queries are $K$ per frame, the learning complexity of the tubelet query grows exponentially to the power of $K$ w.r.t the temporal duration of video clip $T$. To alleviate such an issue, TubeR (Zhao et al., 2022) assumes actor's position changes very slowly over time, thus significantly reducing the complexity (from $\mathcal{O}(K^T)$ to $\mathcal{O}(KT)$). However, this simplification constrains TubeR's ability to effectively learn complex action tubelets, as discussed in the Introduction.

Our work targets complex action tubes. We build ***actor tubelet query***, each of which is supposed to trace a specific actor along time in the input video clip. It applies an Actor Decoder to detect actors on the keyframe (i.e the center frame of the input video clip). Subsequently, we generate actor tubelet queries that leverage the distinctive features of each actor.

#### 3.3.2 ACTOR DECODER

To generate actor tubelet query, we first design an Actor Decoder to detect actors based on the keyframe features $F_{ky} \in \mathbb{R}^{C \times H' \times W'}$. Following DETR (Carion et al., 2020), Actor Decoder learns a set of spatial embeddings $Q_s=\{Q_s^1, ..., Q_s^n, ..., Q_s^N\} \in \mathbb{R}^{N \times C}$ to query persons on the 2D-spatial feature maps $F_{ky}$. With Transformer backbone in Encoder (3.2), the Actor Decoder is structured with $n$ vanilla transformer-decoder blocks (Vaswani et al., 2017). As general, each transformer-decoder

block consists of a self-attention layer (SA), a cross-attention layer (CA), three normalization layers and a feed forward network (FFN). We only illustrate the core attention layers below. The self-attention layer is applied to spatial embeddings $Q_s$ to model the relationships between these spatial queries and generate the spatial queries features $F_s \in \mathbb{R}^{N \times C}$:

$$F_s = \text{SA}(Q_s) = \text{softmax}(\frac{\sigma_q(Q_s) \times \sigma_k(Q_s)^T}{\sqrt{C}}) \times \sigma_v(Q_s) \tag{7}$$

Then, the cross-attention layer decodes the spatial queries features $F_s \in \mathbb{R}^{N \times C}$ from the keyframe features $F_{ky}$ and yields the final features $F_a \in \mathbb{R}^{N \times C}$ for actor detection.

$$F_a = \text{CA}(F_s, F_{ky}) = \text{softmax}(\frac{F_s \times \sigma_k(F_{ky})^T}{\sqrt{C}}) \times \sigma_v(F_{ky}) \tag{8}$$

$\sigma(*)$ is a linear transformation. Two FC layers are respectively applied for actor bounding box regression and classification, to yield actor bounding boxes $B_a \in \mathbb{R}^{N \times 4}$ and actor scores $S_a \in \mathbb{R}^{N \times 2}$.

### 3.3.3 TUBELET QUERY GENERATOR

The Actor Decoder supplies actor location information, which is used to generate actor tubelet queries. Fig 5 delivers a detailed illustration of the Tubelet Query Generator.

**Actor tubelet query.** The final feature $F_a$ from the Actor Decoder is designed for actor detection, but it does not differentiate between individual actors. To distinguish between different individuals, we employ ROI Align (He et al., 2017) to pool actor features $F_{\text{actor}}$ from the keyframe feature $F_{ky} \in \mathbb{R}^{C \times H' \times W'}$ using actor boxes $B_a$ detected from the keyframe. Compared to $F_a$, the lower-level actor feature $F_{\text{actor}} \in \mathbb{R}^{N \times C \times ps \times ps}$ ($ps$ is the spatial size of the pooled feature) is supposed to capture the specific appearance of an actor within the video clip. A linear layer further transforms $F_{\text{actor}}$ into actor queries $A_q \in \mathbb{R}^{N \times C}$. These actor queries are then expanded along the temporal dimension to construct actor tubelet queries $Q_a \in \mathbb{R}^{N \times T \times C}$. Each tubelet query is designed to represent an actor across multiple frames, ensuring consistent identification of the same actor throughout the video clip.

**Temporal compensation.** The above actor tubelet queries $Q_a$ miss temporal information, such as, actors' pose and shape changing along time. Thus, we introduce another concept termed temporal embeddings $Q_t \in \mathbb{R}^{T \times C}$ which is expected to encode the temporal changes to actors. We use same temporal embeddings for $N$ actor tubelet queries $Q_a$. Thus, $Q_t$ is expanded to a temporal compensation features $T_c \in \mathbb{R}^{N \times T \times C}$ and is then added to the actor tubelet queries $Q_a$ when decoding actions.

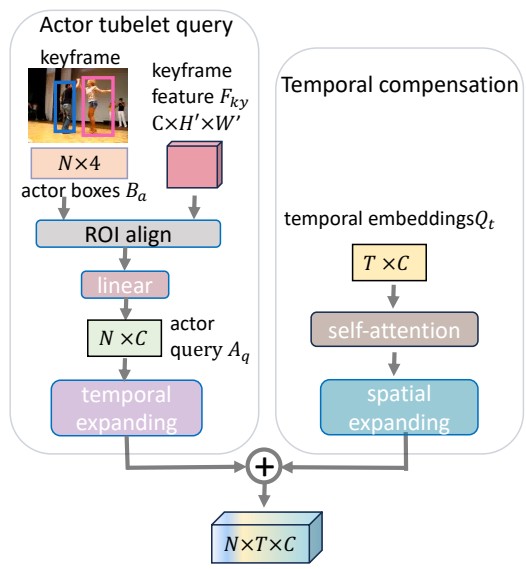

Figure 5: Details of Tubelet Query Generator. It includes the actor tubelet query and the temporal compensation.

### 3.3.4 DISCUSSION

There are three key distinctions between ART and TubeR. **1) Impact**: ART learns actor-related tubelets for complex-shaped action tubes; TubeR, constrained by the position assumption, learns position-related tubelets and struggles to detect complex-shaped tubes. **2) Computation**: ART learns a set of spatial embeddings $Q_s \in \mathbb{R}^{N \times C}$ and temporal embeddings $Q_t \in \mathbb{R}^{T \times C}$. Again, consider that the average number of actor queries are $K$ per frame, the learning complexity is only $\mathcal{O}(K + T)$, which offers a significant reduction compared $\mathcal{O}(KT)$ in TubeR. **3) Structure**: ART is more concise whereas TubeR requires short-term context module and the memory bank (Wu et al., 2019).

## 3.4 ACTION DECODER

Similar to Actor Decoder, Action Decoder consists of $m$ transformer-decoder blocks, each of which contains a self-attention layer (SA) and a cross-attention layer (CA). It decodes action tubelet features $F_{\text{action}} \in \mathbb{R}^{N \times T \times C}$ through the encoded video features $F$ and actor tubelet queries $Q_a$. To capture temporal information, the temporal compensation features $T_c \in \mathbb{R}^{N \times T \times C}$ is added to $Q_a$ in each decoder layer. At last, two FC layers are respectively utilized for regressing bounding boxes on each frame in the tubelet and recognizing actions for each tubelet. We get action tubelet boxes $B_t \in \mathbb{R}^{N \times T \times 4}$ and action probabilities $S_t \in \mathbb{R}^{N \times L}$. $L$ is the number of action categories.

## 3.5 LOSSES

We train the whole system in an end-to-end fashion. We adopt tubelet matching (Zhao et al., 2022) and the total loss is a linear combination of actor detection losses and action detection losses:

$$\begin{aligned}
\mathcal{L} = {} & \lambda_1 \mathcal{L}_{\text{class}}^{\text{actor}} + \lambda_2 \mathcal{L}_{\text{box}}^{\text{actor}} + \lambda_3 \mathcal{L}_{\text{iou}}^{\text{actor}} \\
& + \lambda_1 \mathcal{L}_{\text{class}}^{\text{action}} + \lambda_2 \mathcal{L}_{\text{box}}^{\text{action}} + \lambda_3 \mathcal{L}_{\text{iou}}^{\text{action}},
\end{aligned} \tag{9}$$

The classification loss for actor $\mathcal{L}_{\text{class}}^{\text{actor}}$ is a cross entropy loss and that for action is a focal loss (Lin et al., 2017) . The $\mathcal{L}_{\text{box}}^{*}$ and $\mathcal{L}_{\text{iou}}^{*}$ denote the per-frame bounding box matching error following TubeR. $\lambda_1$, $\lambda_2$ and $\lambda_3$ are weights for classification losses, box regression losses and IoU losses. Empirically, we set $\lambda_1 = 1$, $\lambda_2 = 5$ and $\lambda_3 = 2$.

## 4 EXPERIMENTS

**Datasets.** Our experiments are conducted on three video datasets with tube-level annotations. **JHMDB51-21** (Jhuang et al., 2013) consists of 21 action categories presented in 928 trimmed videos. We report the average results over all three splits. Although action tubes on JHMDB exhibit less variety, it remains a widely-used benchmark for tube-level action detection. **UCF101-24** (Soomro et al., 2012) features 24 sport-related classes distributed across 3,207 untrimmed videos. It contains 20% high complex action tubes as analyzed in Fig 3. We use the revised annotations following (Singh et al., 2017) and report performance on "split-1". **MultiSports** (Li et al., 2021) is a large-scale multi-person dataset for sports actions. It is built on 4 sports classes, collects 3200 video clips, and annotates 37701 action tube instances with 902k bounding boxes. It has well-defined tube-level annotations. And most of the action tubes are complicated. Thus, it is a suitable datasets for validating our method. MultiSports contains 66 fine-grained classes. Following the official evaluation protocol (Li et al., 2021), we only do evaluation on 60 classes.

**Evaluation criteria.** ART is specifically designed for **action tube detection**. Thus, we primarily report video-mAPs. Frame-mAPs are not reported, as they do not directly reflect the effectiveness of ART. For those interested in frame-mAPs on AVA, please refer to the supplementary material.

**Implementation details.** We apply a Transformer backbone Hiera (Ryali et al., 2023) with ViT (Dosovitskiy et al., 2021) in Encoder for a pure transformer system. For fair comparisons to existing methods, we also conduct experiments with ConvNets backbones, including I3D (Carreira & Zisserman, 2017) with VGG (Simonyan & Zisserman, 2015), CSN (Tran et al., 2019) with ResNets (He et al., 2016), which further verify the universality of our design. All backbones are pre-trained on Kinetics-400 (Kay et al., 2017). We set the number of query $N = 10$ on UCF and JHMDB and $N = 20$ on MultiSports. Video clip length is compatible with the used video backbone if not specified. During training, we use the bipartite matching (Georgiev & Lió, 2020) based on the Hungarian algorithm (Kuhn, 1955) between predictions and the ground truth. For data augmentation, each video is resized to 256 pixels and randomly cropped 224 on the short edge, if not specified. We use the AdamW (Loshchilov & Hutter, 2017) optimizer with an initial learning rate $5e - 6$ for the backbone and $1e - 4$ for others. We decrease the learning rate 10 when the validation loss saturates. The weight decay is set to $1e - 8$.

### 4.1 ABLATIONS

We carry on ablation study on UCF, focusing exclusively on RGB input, to highlight the effectiveness of our design using video-mAP@IoU=0.5. We implement a baseline based on position-related tubelet queries, which are utilized in DETR-style detector like TubeR.

|                  | UCF101-24 |      | MultiSports |      |
|------------------|-----------|------|-------------|------|
|                  | High      | Low  | High        | Low  |
| Position-related | 51.1      | 60.8 | 28.5        | 31.3 |
| Actor-related    | 56.8      | 63.3 | 35.1        | 34.8 |
| Δ ↑              | **+5.7**  | +2.5 | **+6.6**    | +3.5 |

Table 1: **Effective for complex action tubes.** Actor-related tubelet performs better than position-related tubelet on all subsets with different levels of complexity, especially biggest gains on high complexity subset.

|                  | s-rate | video-mAP@0.5 |
|------------------|--------|---------------|
| Position-related | 2      | 59.6          |
| Actor-related    | 2      | **60.4**      |
| Position-related | 4      | 56.5          |
| Actor-related    | 4      | **58.5**      |

Table 2: **Effective for large motion.** s-rate means sampling rate. Actor-related tubelet performs better than position-related tubelet at all settings.

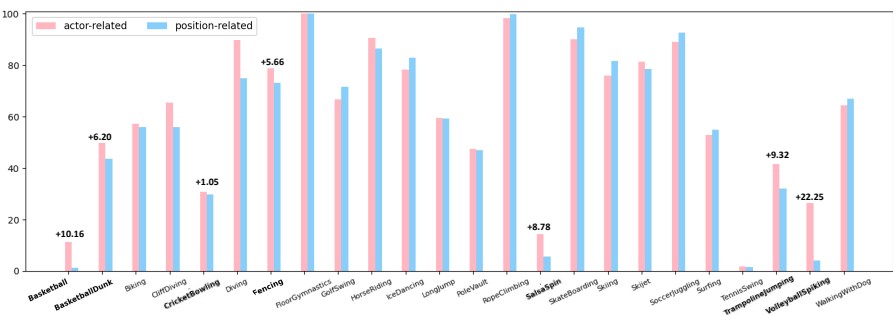

Figure 6: **Effective for scenario with multiple actors.** The actor-related tubelet achieves higher video-mAP on categories with crowded persons.

**Effective for complex-shaped action tubes.** To demonstrate the effectiveness of ART in handling complex action tubes, we divide the UCF and MultiSports datasets into subsets based on high and low complexity scores. Since action tubes in JHMDB exhibit relatively low complexity, we use it as a baseline to establish a threshold for subset division. Seen in Fig 3 (c), we select the complexity score 0.75 as a threshold for dividing subsets, which almost excludes JHMDB from high complexity subset. And we report video-mAP@0.5 on each subset for UCF and MultiSports in Tab 1. Notably, actor-related tubelets show improvements across all settings, with the most significant gains observed on the high-complexity subsets of UCF and MultiSports, achieving increases of +5.7 and +6.6, respectively. This improvement is attributed to actor-related tubelets' ability to track specific actors in complex trajectories. These findings strongly validate our design.

**Effective for large motion.** In Tab 2, we further conduct experiments to assess how well our actor tubelet queries perform in scenarios involving significant motion between frames. We sampled videos at varying rates to simulate different levels of motion. It's important to note that as the sampling rate increases, so does the motion between frames. Actor-related tubelet has +2% gains compared to the position-related tubelet at sampling rate 4, while +0.8% for sample rate 2. These findings suggest that actor-related tubelets are particularly advantageous in cases involving large motions, whereas position-related tubelets perform under the assumption of smaller actor displacements.

**Temporal compensation.** Tab 3 reflects the effectiveness of the temporal compensation module. Incorporating temporal information into actor-related tubelet queries accounts for changes in actors' poses and shapes over time, resulting in a 0.5 improvement in video-mAP.

|        | video-mAP@0.5 |
|--------|---------------|
| w/o tc | 63.7          |
| tc     | **64.2**      |

Table 3: **Temporal compensation** (tc) helps improve video-mAP@0.5.

**Effective for scenario with multiple actors.** From another perspective, we further show per-category video-AP@IoU=0.5 comparisons between the actor-related tubelet and the position-related tubelet in Fig 6. We observe that the actor-related tubelet achieves higher AP scores in categories involving multiple persons, such as an improvement of +10.16 AP for "Basketball" and +22.25 for "VolleyballSpiking". This is because the actor-related tubelet is able to discriminate between actors and form precise tubes.

**Actor decoder vs. Offline person detector.** In Tab 4, we compare our Actor Decoder to an offline person detector, commonly used in most two-stage methods (Feichtenhofer et al., 2019; Ryali et al., 2023). We use Faster RCNN-R50-FPN as the offline person detector. We finetune it for person

| | *Actor* | | | *Action* | |
|---|---|---|---|---|---|
| | train | test | actor-AP@0.5 | strategy | video-mAP@0.5 |
| Two-stage | gt-boxes | faster-rcnn | 73.4 | fixed-pos roi | 18.0 |
| Two-stage | gt-boxes | faster-rcnn | 73.4 | action decoder | 35.7 |
| ART (ours) | actor decoder | actor decoder | 75.0 | fixed-pos roi | 53.2 |
| ART (ours) | actor decoder | actor decoder | 75.0 | action decoder | **64.2** |
| Oracle | gt-boxes | gt-boxes | - | action decoder | 75.3 |

Table 4: **Ablation on Actor Decoder and Action Decoder.** Two-stage models use an offline person detector to localize persons on a keyframe. ARTs apply Actor Decoder. We compare two varieties respectively for two-stage and ART with regard to the strategies for action detection. Oracle represents adopting ground-truth (gt) boxes on a keyframe to obtain actor-related features. Oracle is a reference to show the upper bound of Actor Decoder and should not be directly compared with other results. 'fixed-pos roi' means fixing actor's positions in a video clip and forming a cuboid roi pooling. It shows utilizing both Actor Decoder and Action Decoder achieves the best.

| | *UCF101-24* | | | *JHMDB51-21* | |
|---|---|---|---|---|---|
| | 0.20 | 0.50 | 0.50:0.95 | 0.50 | 0.50:0.95 |
| ACT (Kalogeiton et al., 2017) * | 77.2 | 51.4 | 25.0 | 73.7 | 44.8 |
| TacNet (Song et al., 2019) * | 77.5 | 52.9 | 24.1 | 73.4 | 44.8 |
| TwoinOne (Zhao & Snoek, 2019) * | 78.5 | 50.3 | 24.5 | 74.7 | 45.0 |
| MOC-DLA34 (Li et al., 2020a) * | 82.8 | 53.8 | 28.3 | 77.2 | 59.1 |
| CFAD-I3D (Li et al., 2020b) * | 81.6 | 64.6 | 26.7 | 85.3 | 53.0 |
| TubeR-I3D (Zhao et al., 2022) * | 85.3 | 60.2 | 29.7 | 80.7 | - |
| MOC-DLA34 (Li et al., 2020a) | 78.2 | 50.7 | 26.2 | - | - |
| T-CNN-C3D (Hou et al., 2017) | 47.1 | - | - | - | - |
| TAAD-R50 (Singh et al., 2022) | 79.6 | 52.0 | 23.0 | - | - |
| TubeR-I3D (Zhao et al., 2022) | 82.8 | 57.7 | 28.6 | 78.3 | - |
| **ART**-I3D (ours) | *86.1* | *61.1* | *29.3* | *79.0* | *54.2* |
| TubeR-CSN152 (Zhao et al., 2022) | 83.3 | 58.4 | 28.9 | 82.3 | - |
| **ART**-CSN152 (ours) | *85.6* | *59.4* | *29.1* | *82.8* | *56.4* |
| **ART**-ViT-B (ours) | 89.2 | 64.2 | 32.3 | 87.1 | *61.7* |
| **ART**-ViT-L (ours) | **89.5** | **66.3** | **34.2** | **92.0** | **67.7** |

Table 5: **Comparison on UCF and JHMDB** with video-*mAPs*. ART achieves better results compared to most state-of-arts. * means the method uses both RGB frames and optical flow.

detection on UCF with action box annotations (Notably, an action box must contains an actor). It achieves 73.4 for actor-AP@IoU=0.5. And our Actor Decoder delivers 75.0 for actor-AP@IoU=0.5.

We further analyze the impact of our Actor Decoder on action detection performance. Applying the offline person detector is the so-called Two-stage method (Feichtenhofer et al., 2019). For training Two-stage models, we use ground-truth box annotations like in (Feichtenhofer et al., 2019). And offline person detected boxes are used in test. As seen in Tab 4, comparing Row 2 and Row 4, our Actor Decoder performs much better than the offline person detector for action detection, with the same strategy used for action. Same conclusion is drawn by comparing Row 1 and Row 3. The reason is that the offline person detector may detects persons who are not performing any action, which introduces false alarms. However, our Actor Decoder is trained for actor (persons who are performing actions) detection based on video features. Moreover, we show an Oracle model which replaces our Actor Decoder to ground-truth box annotations for constructing actor tubelet queries. As a reference, the Oracle model supplies an upper bound for our Actor Decoder.

| Model | Tracker | *mAP@0.2* | *mAP@0.5* |
|---|---|---|---|
| SlowFast-R50+Tracks (Singh et al., 2022) | YOLOv5 (Redmon et al., 2016) | 56.3 | 33.0 |
| TAAD-R50+TCN (Singh et al., 2022) | YOLOv5 (Redmon et al., 2016) | **60.6** | **37.0** |
| YOWO (Köpüklü et al., 2019) | ✗ | 12.9 | 9.7 |
| MOC (Li et al., 2020a) | ✗ | 12.9 | 9.7 |
| SlowFast-R50 (Li et al., 2020a) | ✗ | 24.2 | 9.7 |
| SlowFast-R101 (Li et al., 2020a) | ✗ | 28.1 | 8.4 |
| TubeR-R50 (Zhao et al., 2022) | ✗ | 59.4 | 31.7 |
| **ART**-R50 (ours) | ✗ | **62.8** | **36.0** |

Table 6: **Comparisons on MultiSports**. ART perfoms best on MultiSports without an offline tracker.

**Action Decoder vs. Fixed position ROI pooling.** Besides, Tab 4 also presents the effectiveness of our Action Decoder. Comparing Row 1 with Row 2, or Row 3 with Row 4, Action Decoder significantly outperforms fixed-position ROI pooling (expanding detected person boxes on a keyframe along time dimension to form a cuboid to do ROI pooling) (Feichtenhofer et al., 2019) for both two-stage models and ART. Fixed-position ROI pooling works well for action recognition, but largely destroys

tubelet-level box regression. Action Decoder with the help of actor tubelet queries, is able to inquire a specific actor on each frame within a video clip.

## 4.2 MAIN RESULTS

In Table 5, we conduct a comprehensive comparison of ART with state-of-the-art models, focusing on video-mAPs. ART consistently outperforms recent single-stream models on both UCF and JHMDB. In particular, the ART-I3D model achieves higher mAPs than TubeR-I3D across all metrics for UCF, showcasing gains of +3.3 and +3.4 for video-mAP at IoU 0.2 and 0.5, respectively. These equitable comparisons strongly emphasize that the superior performance of our model is attributed to our innovative design, rather than relying on strong backbones. It's worth noting that ART demonstrates moderate improvements (+0.7 and +0.5 respectively for I3D and CSN152) when compared to TubeR on JHMDB for video-mAP at IoU 0.5. This is because action tubes on JHMDB are less complex. They are generally short (no more than 40 frames), featuring a single actor per video, and minimal actor motion over time. ART targets on high complicated tubes. Additionally, our ART model, featuring a Transformer backbone (VIT-L) pre-trained on Kinetics-400, achieves new state-of-the-arts on both datasets, surpassing even performance of two-stream methods.

In Table 6, we compare ART with existing methods on Multi-Sports. For fair comparisons, without an additional tracker, ART demonstrates substantial improvements over YOWO (Köpüklü et al., 2019) and MOC (Li et al., 2020a). It exhibits gains of +3.4 and +4.3, respectively, for video-mAP@0.2 and video-mAP@0.5 compared to TubeR. It is noted that (Singh et al., 2022) utilized a well-built tracker, which definitely helps to localize actors. However, our ART without any bells and whistles even outperforms (Singh et al., 2022) for video-mAP@0.2 and produces comparable result for video-mAP@0.5. This proves the effectiveness of actor-related tubelet design.

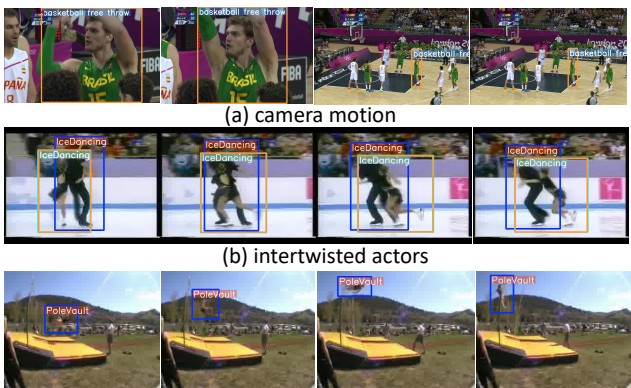

(a) camera motion

(b) intertwisted actors

(c) drastic deformation

Figure 7: Visualization of complex action tubelets on MultiSports and UCF101-24. We use different colors to label different detected tubelets. ART is able to (a) handle camera motion, (b) works well for intertwisted actors and (c) generate tubes with deformable shapes.

**Model and parameter efficiency.** We conduct a fair comparison between ART-I3D and TubeR-I3D in terms of model and parameter efficiency. Using the same input size of $7 \times 224 \times 224$, ART-I3D has 70.8 GFLOPs and 28.3M parameters, whereas TubeR-I3D has 90.1 GFLOPs and 30.3M parameters. Notably, ART requires fewer computations.

## 4.3 VISUALIZATION

We visualize detected complex action tubelets on MultiSports and UFC in Fig. 7. Different colors label different detected action tubelets. (a) well illustrates actor tubelet query is able to handle camera motion. (b) shows a case for intertwisted actors in a video. (c) present tubes with deformable shapes. ART is capable to trace an actor in a tubelet. More visualizations and failure case analysis are in the supplementary material.

## 5 CONCLUSION

We propose Actor-related Tubelets (ART) as an end-to-end solution for complex action tube detection. Unlike existing methods that rely on positional assumptions, ART integrates actor-specific information to generate action tubes, enabling the consistent tracking of the same actor over time. ART not only effectively handles large shifts in actor position but also reduces the complexity of learning. Furthermore, ART demonstrates significant improvements in action detection across multiple datasets, particularly on the complex dataset MultiSports.

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
