# ART: Actor-Related Tubelet for Detecting Complex-shaped Action Tubes Supplementary Material

**Frame-mAP report on AVA.** AVA (Gu et al., 2018) is not the target dataset for our work, as it lacks tube-level annotations. Our ART framework is specifically designed for detecting complex action tubes, and frame-mAP does not adequately capture the effectiveness of ART. Following the main paper, Fig 1 illustrates the cumulative density function of the IoU for ground-truth bounding box pairs taken one second apart, plotted for the training sets of MultiSports, UCF, JHMDB, and AVA. On AVA, 90% of the box pairs have an IoU greater than 0.5, indicating that the motion in AVA is relatively small. However, for those interested in performance on AVA, we provide a comparison with existing methods on AVA 2.2 in Tab 1. Notably, most state-of-the-art methods rely on an offline person detector (typically Faster-RCNN) to first localize actors

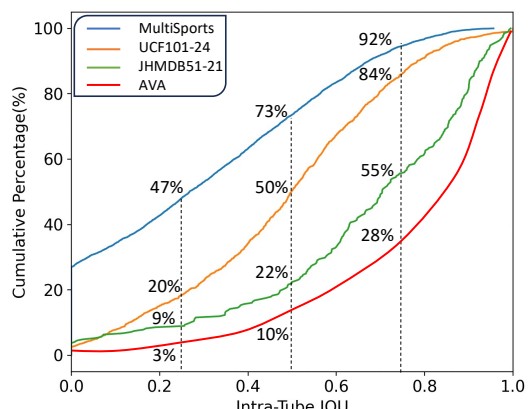

Figure 1: **Cumulative density function of intra-tube IoU** is presented for four action detection datasets: MultiSports, UCF101-24, JHMDB51-21, and AVA. Notably, only 10% of box pairs in AVA exhibit an IoU below 0.5, indicating that 90% of instances in this dataset experience small motion, with bounding boxes overlapping by more than 0.5.

and then focus solely on action recognition. In contrast, our ART method operates end-to-end, simultaneously localizing actors and recognizing their actions. Using only Kinetics-400 pre-trained weights and without incorporating an additional detector, the pure transformer version of ART achieves 40.1 mAP. Although ART is specifically designed for complex-shaped tube detection, its architecture does not compromise performance on actions with small motion trajectories.

**More visualizations.** We present additional action tube detection results on the MultiSports, UCF101-24, and JHMDB51-21 datasets in Fig 2. MultiSports features complex-shaped action tubes, including challenges such as camera motion, deformable shapes, and multiple actors as shown in Fig 2(a). UCF101-24 contains similarly complicated scenarios, such as intertwined actors and multiple actor interactions, as illustrated in Fig 2(b). ART effectively handles these intricate action tubes by leveraging actor information to construct tubelets. As noted in the main paper, JHMDB51-21 (Fig 2(c)) consists of simpler cases, characterized by short-length tubes, single actors, and small motion, as shown in the figure. As expected, ART performs well on this dataset.

**Failure case.** Our ART framework encounters challenges when handling extremely small actors, which complicates the extraction of actor-related information. An example of this issue is illustrated in Fig. 3. In particular, ART occasionally misses bounding boxes within a tube when actors are very tiny. We consider to apply multi-scale technology on both temporal and spatial dimensions to eliminate the issue. We will make it in the future work.

| Model | Detector | Backbone | Pre-train | Inference | *mAP* |
|---|---|---|---|---|---|
| SlowFast (Feichtenhofer et al., 2019) | F-RCNN | R101 | K600 | 6 views | 29.8 |
| ACAR-slowfast (Pan et al., 2021) | F-RCNN | R101 | K600 | 6 views | 33.3 |
| AIA-slowfast (Tang et al., 2020) | F-RCNN | R101 | K700 | 18 views | 32.2 |
| X3D-XL (Feichtenhofer, 2020) | F-RCNN | X3D-XL | K700 | 1 view | 27.4 |
| Unified (Arnab et al., 2021) | F-RCNN | R101 | K400 | 1 view | 28.8 |
| WOO-slowfast (Chen et al., 2021) | ✗ | R101 | K600 | 1 view | 28.3 |
| TubeR-CSN (Zhao et al., 2022) | ✗ | R152 | IG65M | 1 view | 31.1 |
| MViTv1-24 (Fan et al., 2021) | F-RCNN | MViT-B-24 | K600 | 1 views | 28.7 |
| MViTv2-L, $312^2$ (Li et al., 2022) | F-RCNN | MViT-L | IN21K+K700 | 1 views | 34.4 |
| MemViT-24 (Wu et al., 2022) | F-RCNN | MViT-B-24 | K700 | 1 views | 35.4 |
| VideoMAE (Tong et al., 2022) | F-RCNN | ViT-L | K400 | NA | 37.0 |
| **ART**-ViT-L (ours) | ✗ | ViT-L | K400 | 1 view | 38.1 |
| VideoMAE (Tong et al., 2022) | F-RCNN | ViT-H | K400 | NA | 39.5 |
| **ART**-ViT-H (ours) | ✗ | ViT-H | K400 | 1 view | **40.1** |

Table 1: **Comparisons on AVA v2.2** validation set. Detector shows if additional detector is required; IG denotes the IG-65M dataset, SF denotes the slowfast network. Our ART performs best without an offline person detector.

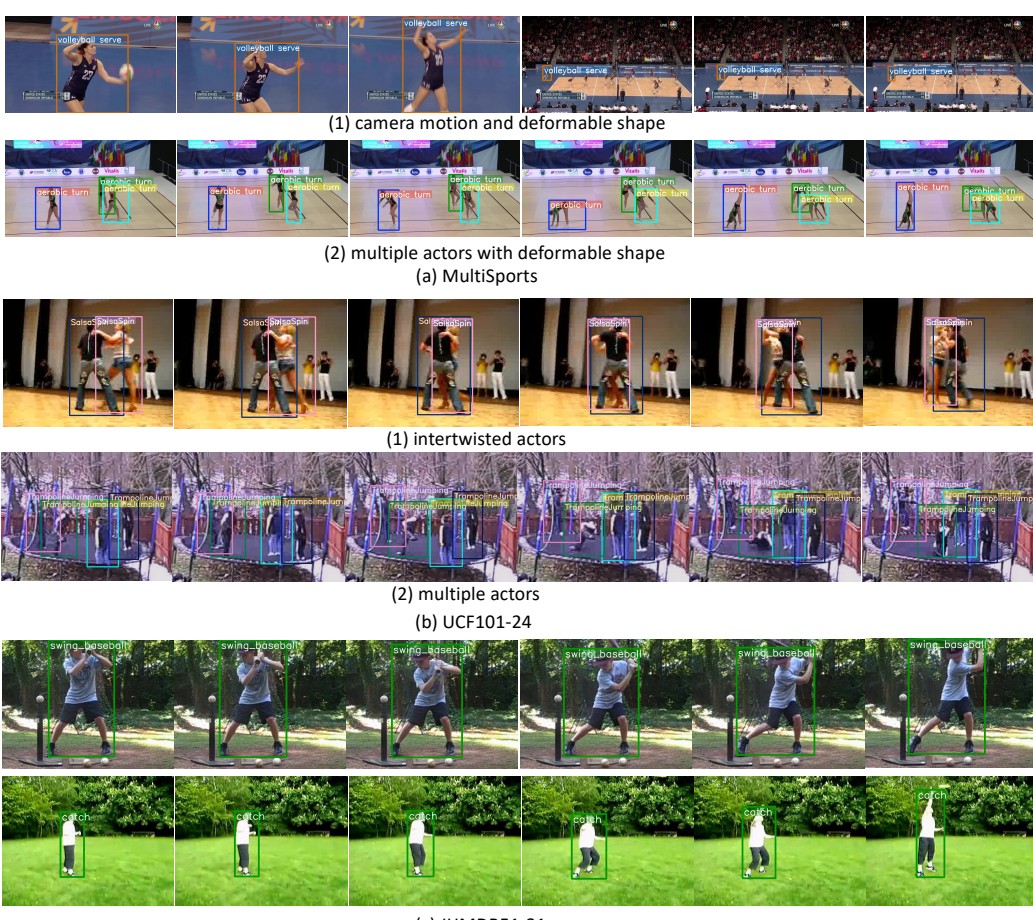

(1) camera motion and deformable shape

(2) multiple actors with deformable shape

(a) MultiSports

(1) intertwisted actors

(2) multiple actors

(b) UCF101-24

(c) JHMDB51-21

Figure 2: **Action tube visualization.** (a) Complex-shaped tubes involving camera motion and multiple actors in MultiSports. (b) Complex-shaped tubes with intertwisted actors and multiple actors in UCF101-24. (c) JHMDB51-21 has tubes characterized with single actor, small motion and short length. ART performs well for various cases.

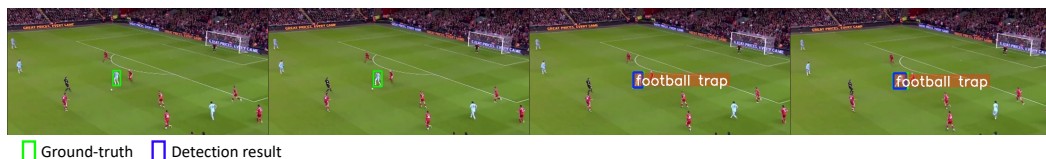

☐ Ground-truth ☐ Detection result

Figure 3: **Failure case**. ART faces challenges when dealing with extremely small actors, as it becomes difficult to incorporate precise actor information necessary for constructing actor-related tubelets.