# OpenReview forum: "ART: Actor-Related Tubelet for Detecting Complex-shaped Action Tubes"
_ICLR.cc/2025/Conference — ICLR 2025 Conference Withdrawn Submission_

### Official Review · Reviewer_r5dE · 2024-10-29

**Soundness:** 3
**Presentation:** 2
**Contribution:** 3
**Rating:** 3
**Confidence:** 4

**Summary:**

This paper introduces an interesting problem on detecting the complex-shaped  action tubes.

The main motivation is that the authors notice that existing methods rely only on the position information to determine the action tubes, lacking the ability to trace the same actor when their movement patterns are complicated.

They introduce actor-related tubelet (ART) which incorporates the actor-specific information when generating the action cubes.

They perform evaluation on 3 benchmark datasets and show that the proposed method achieves better performance compared to existing methods.

**Strengths:**

+ Good and clear motivation, with nice visualisations presented in the paper, eg, Fig. 1

+ The authors conduct some interesting experiments, analysis and comparisons, such as Fig. 3.

+ Experimental results, especially those presented in tables, show that the proposed model achieves better performance. The authors also provide some insights and discussions regarding the proposed model.

**Weaknesses:**

Major:

- Page 1, why “easy actions like brushing hair/teeth … that are spatially stationary or follow predictable patterns”? These motions are just subtle, still considered to be complicated eg the hair movements during brushing process, etc. The descriptions and claims make in the paper needs to be more precise and clear to readers.

- What are the main reasons that cause eg deformable shape in tubes? Would the camera distance, and human subject movement eg far away from camera, or move close camera, can be considered as deformable shape in tubes where human subject sizes changed during the process and eg when the human subject is partially inside the video frame? The authors should make these concepts clear enough.

- Related work section is not polished. It shows significant redundancies especially when the name and citation being repeated in the whole section. The review of closely related works are a bit limited. What are the existing works that doing the action tube detection. The review is also not comprehensive and thorough, a good review should outline the major differences between the method introduced and the existing closely related works.

- Sec. 3, while these metrics and concepts are explained, they are not clear enough to readers. The authors are encouraged to use figures to explain these concepts. Moreover, the method section is not that well-written and well-structured. It is suggested to add a notation section detailing the maths symbols and operations used in the paper to make them clear to readers. The maths symbols presented are also quite raw. What is “*” in Eq. (5), and is it the same as Line 282-283? In Eq. (6), would a weighted sum works?

- Page 5 top, what does it mean “JHMDB is excluded from this plot as it features only a single actor per video”? Does that mean it is unable to visualise it in Fig. 3b?

- The paper is presented in an unnecessarily complicated way, for example, Fig. 4 is quite confusing. The authors should be more specific regarding each maths symbol introduced in texts and figures.  The reviewer gets lost many times and still cannot get the core innovations and insights from the method section, although the motivation is quite interesting.

- Regarding the evaluation metrics, the authors do not clearly explain what is video-mAP, and how this evaluation metric contributes to the performance evaluation of the method.

- Section 4.3, although the authors presented some visualisations, the analysis, evaluations, comparisons and discussions are very limited to 2 sentences. An excellent paper should provide enough insights that are able to deliver powerful and useful information/insights to readers.

Minor:

- Page 6, what is the “ps” value exactly?

- How those hyperparameters are selected, eg, in Eq (9), a reference referring to existing works or how empirically choosing these values would be much clearer and useful.

**Questions:**

Please refer to my main comments above.

**Details Of Ethics Concerns:**

The research experiments conducted in this paper use publicly available datasets. No ethics review needed.

---

### Official Review · Reviewer_RXt2 · 2024-11-03

**Soundness:** 1
**Presentation:** 2
**Contribution:** 1
**Rating:** 3
**Confidence:** 5

**Summary:**

This paper introduces Actor-related Tubelets (ART), an action tublet centric approach for detecting complex-shaped action tubes in videos. ART generates "actor tubelet queries" by extracting ROI-aligned keyframe features with temporal compensation, which are then passed to an Action Decoder to produce action tubelet boxes and scores. The model is evaluated on the MultiSports, UCF101-24, and JHMDB51-21 datasets, where it demonstrates decent results.

**Strengths:**

1. Performance and Analysis:

The reported performance of ART on key datasets is at least decent, and the analysis provided is informative, especially the performance breakdown based on the proposed tube complexity criterion.

2. Metric Innovation:

The proposed metric for measuring tube complexity is a valuable addition.

**Weaknesses:**

1. Questionable Omission of Frame-mAP:

The authors’ decision to exclude frame-mAP results is questionable and limits comparability with prior work in spatio-temporal action recognition. While ART focuses on action tubelets, the tubelet output can certainly be converted to framewise bounding boxes. If the tubelet predictions are indeed high-quality, one would expect the framewise mAP to reflect that as well. Since ART does not propose a new task or benchmark, adhering to conventional evaluation schemes, including frame-mAP, would offer a more transparent assessment of the model’s performance. This addition would also facilitate direct comparison with existing literature, adding further validity to the claims made in the paper.


2. Potentially Misleading Omission of Key Comparisons:

Table 5 primarily compares ART with older works, despite the authors having cited recent influential methods such as “End-to-End Spatio-Temporal Action Localisation with Video Transformers” and “Holistic Interaction Transformer Network for Action Detection.” The omission of these contemporary approaches from the main performance table is problematic, as it may obscure ART's comparative impact. Including these methods would present a more comprehensive and honest performance evaluation.

3. Moderate Impact of Architectural Changes:

While ART introduces an architectural modification that addresses positional constraints, this adaptation alone may not be groundbreaking. ART’s performance gains, while respectable, do not demonstrate a transformative leap, especially if compared directly to the omitted recent works. The improvements seen with ART suggest more of an incremental contribution rather than a paradigm shift in action tube detection.

**Questions:**

Please see weakness section. Not that important but too many underbars in Sec 3.1 equations seem somewhat ackward.

---

### Official Review · Reviewer_avT7 · 2024-11-03

**Soundness:** 2
**Presentation:** 2
**Contribution:** 2
**Rating:** 5
**Confidence:** 4

**Summary:**

This paper introduces actor-related tubelet (ART) for spatial-temporal action localization. It aim to improve the performance in complex scenes. The framework is based on transformer architectures.  To distinguish different actors, the authors employ ROI align to extract actor features using key frame features from the decoder and extend these features temporally to build actor-related queries. The proposed method obtained competitive results on JHMDB51-21, UCF101-24, and MultiSports.

**Strengths:**

1) Different from position-based methods, the proposed ART leverages actor features to distinguish different actors in complex scenes
2) The proposed method shows significant improvements on complex activities.

**Weaknesses:**

1) Notations in Section 3.1 are confusing, variable names and indexes are tangled together.
2) The loss function in Eq. 9 is incorrect, please proofread the equations.
3) Simple action with large motion can also lead to high complexity in Eq. 2
4) The key of the method is actor tubelet query, but there are few discussion of actor-related feature except using the ROI align method in (He et al., 2017)
5) In Figure 6, the performance of many actions using actor-related methods are worse that the ones using position-related methods. And which method is used for the blue bar?

**Questions:**

1) Previous also use DETR to detect actors for generating queries, why they cannot capture actor related information?

**Details Of Ethics Concerns:**

No concerns

---

### Official Review · Reviewer_gELM · 2024-11-04

**Soundness:** 2
**Presentation:** 2
**Contribution:** 2
**Rating:** 3
**Confidence:** 4

**Summary:**

The paper proposed an approach to solve action detection via tubelet based approach. The paper proposes a metric to compute the complexity of the datasets based on tubelet IoUs. Tubelet query Generator helps for actor tubelet queries to boost the performance.

**Strengths:**

The proposed approach outperforms previous approaches on all datasets.

**Weaknesses:**

Novelty -
- (Section 1 Introduction - Lines: 98-102) In the current settings, precise information about the actor location is provided by analyzing the video beforehand. What does this mean? Does the model know the actor locations in all the keyframes? These settings are not consistent with previous approaches [TubeR, EVAD, STAR, BMViT]. Are the previous approaches evaluated in similar settings - where the actor/action locations are provided and then the model is trained and evaluated? How would the model work without these annotations provided beforehand?
- (Section 3.1 Tube shape Complexity) How is the proposed metric different from [1]? Is it a direct extension from [1]? Can authors please elaborate on this?
- (Section 3.3 Actor-Related Tubelet) The proposed approach looks very similar to TubeR each module. I understood the difference between position vs actor against TubeR. Tubelet Query Generator module - It’s not clear how it is different from TubeR’s Tubelet attention module. Can authors please clarify? Also, the complexity analysis is also not clear. Looks like TubeR should have the similar complexity.

(Results & Ablations)
- Ablation Study - There is only one ablation study on temporal compensation for the whole model with just one model and one dataset. Can authors please provide more ablations for different datasets or with different models? The proposed module shows a very minor performance boost for 0.5%. It will be helpful to see if this is the same pattern with other models/datasets.
- Is the results on AVA comparable since the paper is using a better visual backbone?  Without ablations, it is hard to prove that it’s not just a good performance due to better backbone. AVA is a very difficult dataset where an actor comes and goes out of the scene. How are the tubelet of an actor maintained across time?
- (Sec 4.1 - Scenario with Multiple Actors) Does knowing the location of the bounding box helps in performance boost for difficult classes such as Basketball, Basketball Dunk, etc.? Is there any analysis of how prior knowledge helps?
- UCF and JHMDB don't have the issue of actors appearing and disappearing from the video. Do Multisports have this issue? If yes, how is the paper handling the issue that the same tubelet id is assigned similar to any offline tracker?
- (Section 4.1 Actor decoder vs Offline person detector) Does the ablation study show the detector+tracker with some base ReID model to track appearance features? Without a ReID model, the tracker mixes up IDs and leads to false tubelet generation because of ID switching.

**Questions:**

- Could the approach of the tubelet query generator be applied to other works? If so, how? If not, why not? The module looks pretty flexible to be adaptable for video action detection modules.
- Can authors also please show or share f-mAP results on UCF, JHMDB, and Multisports? It will be helpful to compare it to recent works that use frame-based approaches.

Please look into the weakness section for other queries. The weaknesses are listed in priority from high to low.

---

### Note · Authors · 2024-11-28

I have read and agree with the venue's withdrawal policy on behalf of myself and my co-authors.